# New Evaluation of Small Farms: Implication for an Analysis of Food Security

**Zahra Ardakani \*, Fabio Bartolini**  **and Gianluca Brunori**

Department of Agricultural, Food and Environmental Economics, University of Pisa, 56124 Pisa, Italy;
fabio.bartolini@unipi.it (F.B.); gianluca.brunori@unipi.it (G.B.)
**\*** Correspondence: zahra.ardakani@gmail.com; Tel.: +39-346-503-0931

**Abstract:** Farm structure is a multi-dimensional concept that can be measured through different criteria. Meanwhile, farm structure has been identified to discern small farms or well-endowed farms from the other farms. Distinguishing and identifying these two groups have practical implications for understanding the dynamics in rural areas and the effectiveness of target measures in these categories. The existing literature lacks a better definition of small farms based on the different criteria used. In this paper, we have developed composite indicators to apply to the concept of farm structure to re-define small farms and discover their role in achieving food security in Europe. By clustering countries using the composite indicator of farm structure, we estimate that more than 80 percent of food across Europe is produced by multi-criteria small and medium farms, but the partial productivities of agricultural land and labor in these countries that have the majority of multi-criteria small and medium farms are much lower than the large ones. Then, an estimate of a spatial econometric regression model was done to recognize how farm structure, a representative of farm size, can affect food availability, which is representative of food security. The results show that improving the structure of farms in a country not only improves its food security but also improves its neighbors' food security. Hence, improving the structure of multi-criteria small farms is a major part of the solution to improve and achieve food security. Recognizing and understanding the diversity of multi-criteria small farms by considering the specific products and countries is necessary for designing appropriate innovations and policies for supporting more productive multi-criteria small farms.

**Keywords:** multi-criteria small farms; food security; partial productivity; composite indicators; spatial econometric

## 1. Introduction

By theoretical definition, a farm is an economic unit of agricultural production under single management that includes all retained livestock and all land utilized for agricultural production purposes [1,2] without regard to title, legal form, or size [1]. Alternatively, a farm is defined as a civil or juridical person who makes the major decisions regarding the use of resources and exercises management control over the agricultural holding operations [1]. A technical-economic unit under single management engaged in agricultural production is the definition of a farm, according to the Farm Structure Survey carried out by all European Union (EU) Member States.

Based on these definitions and the accuracy of the Farm Structure Surveys in the agricultural censuses [3], it is estimated that there are approximately 570 million farms in the world [4]. The total number of small farms ranked by landmass within this group exceeds 500 million farms [5] that produce more than 80 percent of the food and manage approximately 80 percent of the agricultural land [4,5].

So far, small farms are frequently defined by physical size; and farms are often considered small when they are less than 1 or 2 hectares. According to the agricultural censuses data from a large sample of countries, 72 percent of farms are less than 1 hectare, and 12 percent are between 1 and 2 hectares [4]. Small farms, based on landmass, play an important role in farming in the European Union too [6]. Of the 12 million working farms in Europe, more than 6 million of them are less than 2 hectares; and more than 8 million are less than 5 hectares (69 percent of farms) [6,7].

Agricultural economists and development specialists largely agree that investing in agriculture is an effective strategy for reducing hunger [8]. Hence, small farms based on landmass are central to the growth of sustainable productivity in agriculture [9]. This is crucial, with the available natural resource, for producing more food to meet growing demand [10] and for enhancing food security [11]. A new understanding of the importance of small farms and the contribution they make to food security has elicited renewed attention in both developed and developing countries to discover the various roles of these small farms in the food systems [10]. Nevertheless, there is no common description of small farms.

The size of farms across countries and over time depends on complex factors [10,12] that can be measured in various ways [13]. Although the size of farms is a debatable proxy [14] and a common topic of debate among agricultural development specialists [15], to date, farms are frequently defined only in terms of physical size or land, the most easily comparable criterion [10]. However, land is not the only criterion for measuring the size of farms and for defining small farms.

A farm is small because of scarce resources endowment [16]. Small and large concepts are relative depending on the context [8,10] and vary widely from country to country with considerable variations in approaches to measuring [8]. To design effectively, it is necessary to have a picture of some of the key features that define small farms. Thus, the choice of criteria used to measure size will be the first question to answer. The fundamental driving forces of the structure of farms [15] can be approximately clustered. The key drivers of farm structure are the utilized agricultural area or arable land, the standard output of an agricultural product (which is an average monetary value of the agricultural output at farm-gate price per hectare), the farm labor force who has worked on the farm, and the total number of livestock units as reported in the Farm Structure Survey of the European Union. Consequently, small versus large are relative and depend on the choice of criteria used to measure size [10].

Because different characteristics in the structure of farms are used to determine its size, these characteristics to describe small farms may elicit different results [17]. In this paper, we re-define small farms as a multi-criteria concept [18] through a composite indicator of farm structure, which can cover different criteria [19]. Therefore, this paper proposes a new composite indicator to make a single measure of farm structure, using the features that affect the structure of farms. This measurement will document the distribution of the countries that have multi-criteria small farms in Europe and that understood the role of these countries with multi-criteria small farms in producing food and reaching food security. Then, the development of such an indicator will enable an understanding of changes in the agricultural system and will model the impact of external factors, including new policies and innovations. Moreover, we analyze the composite indicator of farm structure in a spatial econometric regression model to discover how differences in the farm structure may affect food security.

This paper is structured as follows. Section 2 presents methods for concluding the statistical analysis of composite indicators, econometric analysis of spatial panel models, and describes the data. In Section 3, we deliver the results. Section 4 explains our results. The final section concludes the study.

## 2. Materials and Methods

### 2.1. Composite Indicators

A composite indicator, which has no common meaningful unit of measurement [20], is a mathematical or computational combination model of a set of indicators [20,21] that represent different

dimensions of a concept where description is the objective of the analysis [22]. Therefore, a composite indicator is formed when individual indicators are compiled into a single index that is based on an underlying model of the multi-dimensional concept that is being measured. With the scope and usage of composite indicators, they reflect multi-dimensional issues, assess the progress of entities over time, provide benchmarking, and comprehensively rank entities of the phenomena being measured [23,24]. Quantitative composite indicators are useful for comparing and objectifying large amounts of international data and international comparative studies to illustrate how large and complex patterns can be objectified into easily understood formats as composite indicators [25] in various performance and policy areas [26], in several fields, such as environment, economy, society, or technological development [27,28].

Because the size of farms is a debatable proxy, and relying on one proxy to define the size of farms can be misleading, the aim of this paper is to contribute to the methodological foundation in building a composite indicator of farm structure, a representative of farm size, which currently has not been done. Table 1 reports the statistical steps in the construction of a composite indicator. Then a cluster analysis will be applied to the composite indicator of farm structure to categorize the countries and find the role of multi-criteria small farms.

**Table 1.** Statistical steps in the construction of a composite indicator; Source: [20,22].

| Steps | Description |
|:-----:|:-----------:|
| 1 | The multi-dimensional phenomenon is defined. |
| 2 | The related indicators will be identified. |
| 3 | The initial high-quality data available for all the indicators will be collected. |
| 4 | The data will be converted to logarithm to avoid skewed distributions and follow more closely to the normal distribution. |
| 5 | The data will be normalized to convert the different indicators into the same unit. [1] |
| 6 | The data will be weighted to assign the importance of indicators. [1] |
| 7 | The normalized weighted indicators will be aggregated to build the final index. [1] |

[1] Three last steps are subjective and can be defined by researchers according to the issue.

## 2.2. Spatial Regression

Then to recognize how the composite indicator of farm structure, which has been built into the study, can affect food security, a spatial panel econometric regression model [18] has been estimated. Spatial panels typically refer to the data containing time-series observations of many spatial units (zip codes, municipalities, regions, states, jurisdictions, countries, etc.). The main reason to incorporate spatial components in the economic models is to control for spatial dependence; this can be introduced into linear regression models in two ways: Spatial Lag Model (SLM) and Spatial Error Model SEM).

The spatial lag model assumes that the dependent variable depends on the dependent variable observed in the neighboring units and on a set of observed local characteristics. The spatial error model, on the other hand, theorizes that the dependent variable depends on a set of observed local characteristics and that the error terms are correlated across space [29]. Ignoring spatial effects in a regression may lead to biased and inconsistent estimates of the model parameters when a spatial lag is omitted or to inefficient estimates and biased inference when a spatial error is omitted [30]. The application of a spatial econometric regression model will inevitably involve defining a particular spatial weight matrix, which reflects a geographic relationship between regions [31].

The spatial lag models (SLM) and spatial error model (SEM) are specified as follows, respectively, in formula (1) and formula (2):

$$y = \rho w_y + x\beta + \epsilon \tag{1}$$

$$y = x\beta + \epsilon \, (1 - \lambda w_\epsilon)^{-1} \mu \tag{2}$$

where $y$ and $x$ show the dependent and independent variables. $\beta$ is the corresponding regression coefficient, $\varepsilon$ is the model error term and $\mu$ is the normally distributed random error. $\rho$ is a spatial parameter that indicates the spatial extent of interaction between geographic locations. $\lambda$ is another spatial parameter that indicates the intensity of the spatial correlation between the regression residuals. If $\rho$ and $\lambda$ are zero, it indicates there are no spatial effects. $W$ is $n$ by the n-spatial weight matrix that defines a spatial neighborhood.

In the empirical model of the study, concerning the nature of the dataset, which has geographical components (countries), we have examined the existing spatial dependence between the data and then try to estimate a spatial econometric regression model. The dependent and independent variables located in the model are constructed composite indicators.

The main independent variable is a composite indicator of farm structure, which is a representative of farm size as an internal factor that affects food systems and the most important independent variable in the model. The second independent variable of the model is a composite indicator of climatic features that have been considered as an environmental factor, which can affect food systems, and the third independent variable in the empirical model is a composite indicator of economic conditions as an external factor that affects food systems. A composite indicator of food availability as a representative of food security has been considered as the dependent variable in the empirical econometric regression model since food security is the final and main goal of the food systems. The single indicators, which have been used in constructing the above composite indicators, are reported in Table 2 in the data section.

### 2.3. Data

The indicators chosen to build the composite indicator of food availability [32], a representative of food security include average dietary energy supply adequacy (percent), an average value of food production (dollars), a share of dietary energy supply derived from cereals, roots, and tubers (percent), average protein supply (grams) and average supply of protein of animal origin (grams). The data of the composite indicator of farm structure, a representative of farm size, come from the Farm Structure Survey of the agricultural censuses [7]. These indicators are an average area of holdings (hectares), average livestock units of holdings (livestock units), an average labor force of holdings (annual work units) and average standard output of holdings (euros). The key indicators for the composite indicator of climatic features, including temperature (centigrade) and rainfall (millimeter) [33], are available on the World Bank website. Lastly, the indicators of economic conditions are gross domestic product per capita (dollars) and an index of domestic food price levels [32]. A descriptive statistic of the dataset is reported in Table 2.

**Table 2.** Descriptive statistics of the data used in the study (EU+EFTA, 2010); Source: [7,34,35] and own calculations; n = 128.

| Indicators | Mean | Std. Dev. | Min | Max |
|---|---|---|---|---|
| Food availability indicators: | | | | |
| Average dietary energy supply adequacy (%) | 132.31 | 10.25 | 104 | 149 |
| Average value of food production ($) | 444.88 | 203.30 | 179 | 1089 |
| Share of dietary energy supply derived from cereals, roots, and tubers (%) | 32.06 | 5.25 | 23 | 45 |
| Average protein supply (gr) | 102.94 | 12.78 | 74 | 132 |
| Average supply of protein of animal origin (gr) | 60.53 | 11.63 | 36 | 96 |
| Farm structure indicators: | | | | |
| Area of holdings (ha) | 49.54 | 108.27 | 0.91 | 616.09 |
| Livestock units of holdings (LSU) | 30.20 | 32.58 | 1.41 | 116.85 |
| Labor force of holdings (AWU) | 1.24 | 0.80 | 0.39 | 4.72 |
| Standard output of holdings (€) | 61.64 | 66.71 | 2.60 | 261.75 |
| Climatic features indicators: | | | | |
| Temperature © | 9.04 | 4.83 | 0.04 | 20.72 |
| Rainfall (mm) | 78.11 | 24.10 | 33.4 | 132.41 |
| Economic conditions indicators: | | | | |
| Gross domestic product per capita ($) | 34335.28 | 15535.01 | 13785.4 | 90790.8 |
| Domestic food price level index | 2.20 | 0.94 | 1.20 | 5.62 |

The study area includes 32 countries encompasses 28 European countries (EU) and 4 countries in the European Free Trade Association (EFTA) including Belgium, Bulgaria, Czech Republic, Denmark, Germany, Estonia, Ireland, Greece, Spain, France, Croatia, Italy, Cyprus, Latvia, Lithuania, Luxembourg, Hungary, Malta, Netherlands, Austria, Poland, Portugal, Romania, Slovenia, Slovakia, Finland, Sweden, United Kingdom, Iceland, Norway, Switzerland, and Montenegro. According to the data availability, the years of 2005, 2007, 2010, and 2013 have been considered in the analysis.

## 3. Results

The study aims to re-define small farms, assess the countries with multi-criteria small farms, specify the role of multi-criteria small farms in producing food, and find the impact of farm structure on food availability, a representative of food security, across European countries. The results are given in the following three subsections.

### 3.1. Design of Composite Indicators

To reach the first goal of the study, i.e., finding the role of multi-criteria small farms in producing food, the analysis of composite indicators has been developed. In the construction of composite indicators (Table 1), as the last three steps are subjective, in this study, we have normalized the data through a vector normalization to avoid receiving minus numbers (step 5). Then, in step six, we have used factor analysis, as a statistical approach and most preferred [23] to weigh the indicators. The obtained weights have been reported in Table A1 of the Appendix A. In the last step of constructing the composite indicators, the normalized weighted indicators are aggregated to build the final index. We have aggregated the indicators of farm structure, a representative of farm size, by applying output-input ratio (the aggregated outputs divided to the aggregated inputs) or partial productivity conception [36]. Partial productivity is a very old concept, which is defined as the ability of a unit of input to produce a given level of output [37]. Therefore, measuring the partial agricultural land and labor productivities discusses to measure the contribution of each individual production factor to the total agricultural output [38]. The other three composite indicators of food availability, climatic features, and economic conditions have been aggregated through an arithmetic average. The order of the countries through the composite indicator of farm structure, the main debatable composite indicator in the study, has been reported in Table A2 of Appendix A.

### 3.2. Analysis and Definition of Clusters

Then, we have applied a cluster analysis to define a statistical threshold [39] for the composite indicator of farm structure, which has been measured in this study. Through drawing a Dendrogram of the composite indicator of farm structure, the existence of three clusters in the dataset has been identified. Therefore, the countries have been categorized into three clusters using the k-means method. There are 12 countries in cluster one, 14 countries in cluster two and, 6 countries in cluster three (Table 3). As we have used partial productivity concept in the building of the composite indicator of farm structure, the countries in the same clusters seem to be similar in partial agricultural land and labor productivities.

**Table 3.** Defining and re-naming the three clusters of European countries through the composite indicator of farm structure; Source: own calculations.

| Cluster 1: Low-Low Partial Productivities Multi-Criteria Small Farms | Cluster 2: Low-Moderate Partial Productivities Multi-Criteria Medium Farms | Cluster 3: High-High Partial Productivities Multi-Criteria Large Farms |
|---|---|---|
| Greece | Austria | Belgium |
| Hungary | Luxembourg | Netherlands |
| Estonia | Italy | Switzerland |
| Slovakia | Germany | Norway |
| Czech Republic | France | Denmark |
| Iceland | Cyprus | Slovenia |
| Malta | Ireland | |
| Lithuania | Croatia | |
| Latvia | Spain | |
| Bulgaria | United Kingdom | |
| Montenegro | Poland | |
| Romania | Finland | |
| | Sweden | |
| | Portugal | |

To confirm the results of the cluster analysis and find similarities among the countries in the same clusters, we have estimated the partial agricultural land and labor productivities of the countries in the three clusters. Figures 1 and 2 show the results. The average of the partial productivity of agricultural land in cluster 1 is around 4.01 tons per hectares (Figure 1) and the average of partial productivity of agricultural labor in this cluster is around 53.43 tons per annual work units (Figure 2). For cluster 2, the average partial productivity of agricultural land has been obtained at approximately 4.51 tons per hectare (Figure 1), and the average of the partial productivity of agricultural labor is approximately 115.93 tons per annual work unit (Figure 2). Finally, in the last cluster, we have found an average of partial agricultural land productivity of 9.27 tons per hectare (Figure 1) and an average of partial agricultural labor productivity of 184.71 tons per annual work units (Figure 2).

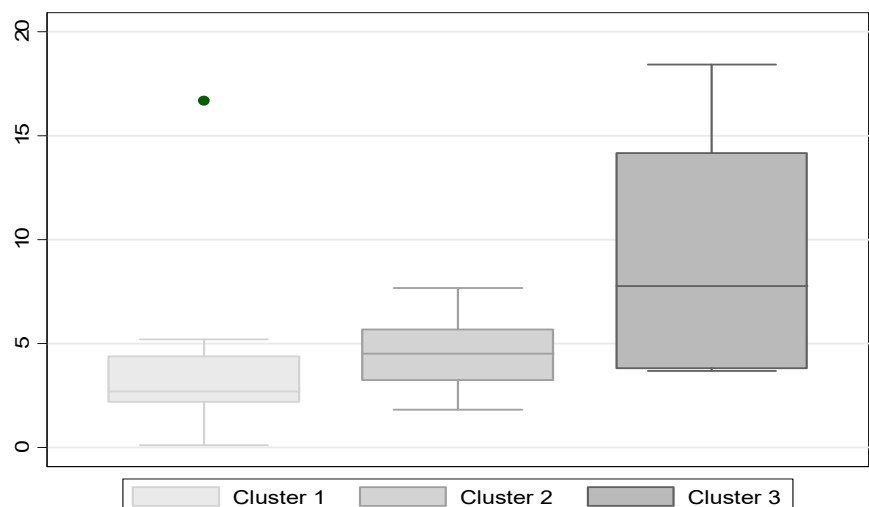

**Figure 1.** Partial Productivity of land in the three clusters across European countries; Source: [34] and own calculations.

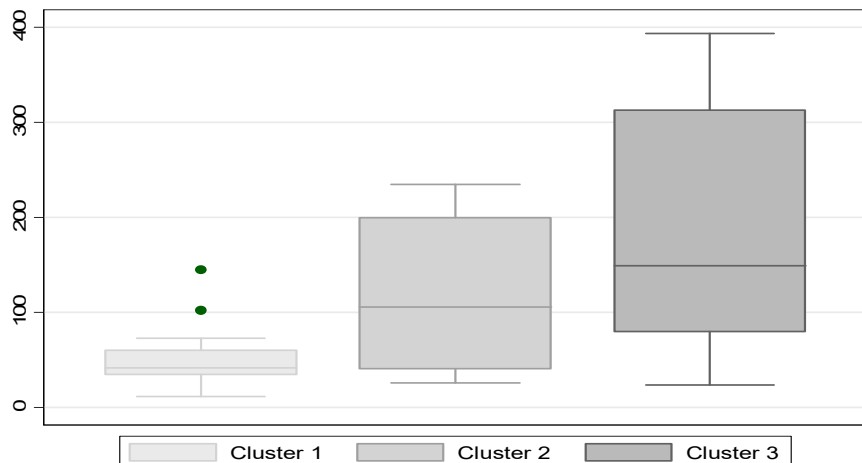

**Figure 2.** Partial Productivity of labor in the three clusters across European countries; Source: [34] and own calculations.

Thus, by using the concept of partial productivity in the clusters, we have re-named the three clusters as follows: cluster 1 can be defined as a cluster with the majority of multi-criteria small farms (small: low-low partial productivities), which has low partial productivity in land and low partial productivity in labor. Cluster 2 can be re-named as a cluster with the majority of multi-criteria medium farms (medium: low-moderate partial productivities), which has low partial productivity in land and moderate partial productivity in labor. Cluster 3 can be re-named as a cluster with most of the multi-criteria large farms (large: high-high partial productivities), which has high partial productivity in land and high partial productivity in labor. Table 3 shows the clustering and defining European countries as multi-criteria small, medium and large farms.

The distribution of farms according to the size that has been measured in this study, has been presented in Figure 3. According to this figure, more than 80 percent of the farms that are producing food in Europe are multi-criteria small and medium, and only 15 percent of the farms can be considered multi-criteria large.

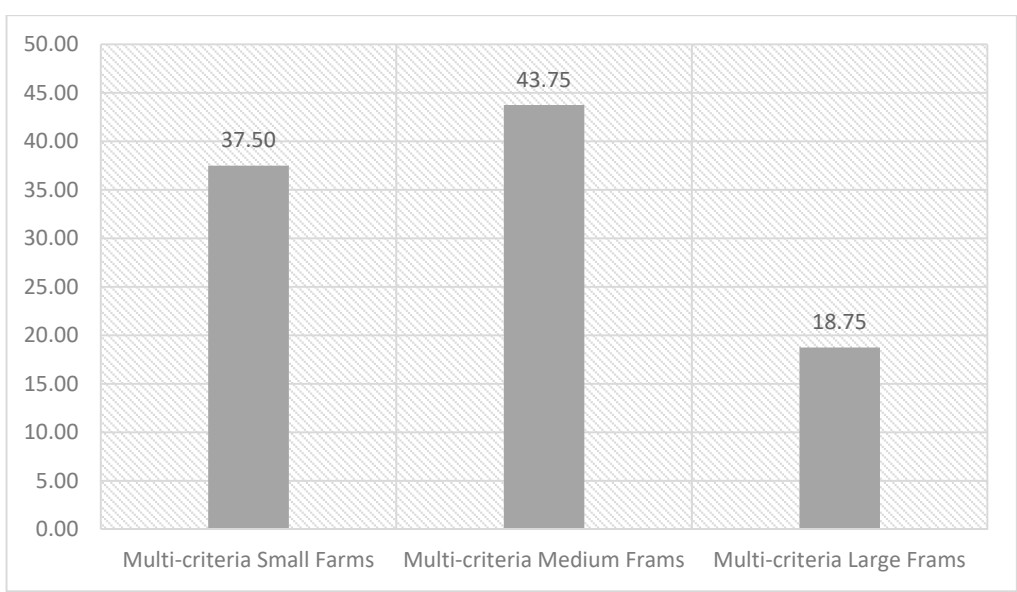

**Figure 3.** Distribution of farms across European countries (percentage); Source: own calculations.

A share of multi-criteria small, multi-criteria medium, and multi-criteria large farms in the production of primary crops and livestock and the share of them in using two important inputs in

farming, i.e., agricultural land and labor have been reported in Table 4. As this table shows, the countries with the majority of multi-criteria small and multi-criteria medium farms are producing around 89 percent of food across Europe. They have got around 95 percent of agricultural land and around 95 percent of the labor force for producing this amount of food. On the other hand, the countries with the majority of multi-criteria large farms, through 5 percent of the land and around 5 percent of the labor force is producing around 10 percent of food in Europe. According to these findings, even though multi-criteria small farms have an important role in producing food in Europe by a production of 89 percent of the food, they are using 95 percent of agricultural land and 95 percent of the agricultural labor for this. Therefore, their food production shows very low partial productivities in agricultural land and agricultural labor force compared to the larger ones.

**Table 4.** Share of the three clusters in (primary) production, land, and labor across Europe; Source: [34] and own calculations.

| Clusters | Production (Million Tons) | | Land (Million Hectares) | | Labor (Million Annual Work Units) | |
|---|---|---|---|---|---|---|
| | Value | Share | Value | Share | Value | Share |
| Multi-criteria Small | 120.44 | 14.34 | 38.63 | 21.71 | 3.34 | 33.02 |
| Multi-criteria Medium | 631.26 | 75.17 | 130.94 | 73.57 | 6.30 | 62.11 |
| Small and Medium | 751.70 | 89.52 | 169.57 | 95.27 | 9.65 | 95.12 |
| Multi-criteria Large | 88.02 | 10.48 | 8.41 | 4.73 | 0.49 | 4.88 |
| Total | 839.72 | 100 | 177.98 | 100 | 10.14 | 100 |

*3.3. Regressions*

Then, by applying a spatial econometric regression model, we have estimated the effect of farm structure on food availability, a representative of food security. The spatial panel models, the spatial lag model, and the spatial error model have been measured with regard to the micro panel dataset available. To start the procedures of specifying the best model for finding the effect of farm structure on food security, we have done the Chow test to recognize the existence of a pool or panel structure in our dataset. According to the result of the Chow test ($F_c = 1.03 < F_t = 2.46$), the null hypothesis was accepted, and we have found an existing pool structure in our micro-panel sample dataset. To have a base to start the procedures [30], a Pool-OLS model (Table 5) has been estimated. After running the Pool-OLS model to test for a spatial correlation in the model, the spatial diagnostics [30] reported in Table 6 have been done. The significance of Moran's I test (that is applied to the residuals of the OLS regression) shows the existence of a spatial effect in the dataset. While the Moran's I test identifies the existence of spatial dependence, it does not specify and provide the underlying spatial process. Multiplier Lagrange tests are more specific and provide a basis for choosing between spatial lag and spatial error specifications. To choose between a spatial lag model and spatial error model, we have decided through the Multiplier Lagrange and Robust Multiplier Lagrange tests. As the Multiplier Lagrange test of the spatial lag model is significant and the Robust Multiplier Lagrange confirms it, we have chosen the spatial lag model to estimate the effect of farm structure on food security.

**Table 5.** The estimated results of Pooled-OLS regression; Source: own calculations.

| Food Availability | Coefficients | *p*-Values | Number of Observations = 112 |
|---|---|---|---|
| Farm structure | 0.002 | 0.013 | F (3, 108) = 12.16 |
| Climatic features | −0.032 | 0.089 | *p*-value > F = 0.000 |
| Economic conditions | 0.210 | 0.000 | *R*-squared = 0.252 |
| Intercept | 0.057 | 0.006 | Adjusted R-squared = 0.231 |

**Table 6.** Diagnostic tests for spatial dependence in OLS regression; Source: own calculations.

| Test | Statistics | *p*-Value |
|---|---|---|
| Moran's I | 2.432 | 0.015 |
| Spatial error: | | |
| Lagrange multiplier | 0.322 | 0.570 |
| Robust Lagrange multiplier | 4.140 | 0.042 |
| Spatial lag: | | |
| Lagrange multiplier | 3.496 | 0.062 |
| Robust Lagrange multiplier | 7.313 | 0.007 |

The results of the spatial lag model have been reported in Table 7. All of the independent variables are significant and they have expected signs. As we mentioned before, the dependent variable is a composite indicator of food availability as a representative of food security (the main goal of food systems). The explanatory variables are a composite indicator of farm structure as an internal factor that affects food systems, a composite indicator of climatic features as an environmental factor that affects food systems, and a composite indicator of economic conditions as an external factor that affects food systems. The spatial lag model explains that every improvement in farm structure in each country will improve food availability and then food security in that country directly and in neighboring countries indirectly. For example, if the structure of farms in Italy changes to make farms more productive, not only will the level of food security in Italy increase but also the level of food security of its neighbors, like France, will increase too. The impact of climatic features on food availability is inconclusive and unpredictable. It can increase food production and then food availability if the right amount of rain and temperature happens at the right time, and vice versa. In this study, climatic features negatively impacted the food availability of a country and its neighbors. Lastly, the impact of economic conditions (gross domestic product per capita and domestic food price level index) on food availability is positive on the supply side. So, improving economic conditions in a country will also improve the food availability and security of the country and its neighbors.

**Table 7.** The estimated results of a spatial lag model regression; Source: own calculations.

| Food Availability | Coefficients | Z-Value | *p*-Values | Number of Observations = 112 |
|---|---|---|---|---|
| Farm structure | 0.001 | 1.98 | 0.048 | Log likelihood = 388.21 |
| Climatic features | −0.033 | −1.85 | 0.064 | AIC = −764.42 |
| Economic conditions | 0.205 | 5.04 | 0.000 | BIC = −748.11 |
| Intercept | −0.012 | −0.25 | 0.804 | *p* = 0.473 |

## 4. Discussion

Agricultural development, especially increasing the partial productivities of agricultural land and labor [40,41], has been increasingly noticed over recent decades [42]. Investing in agriculture is an effective tool [8] for increasing agricultural economic growth [43], reducing poverty, and improving food security [8,43] dependent on how it is implemented [43,44]. The nature of food security is complex and multi-dimensional, and food production is the first dimension of food security [45,46]; food availability is the primary responsibility of agriculture and farming [47]. Therefore, food availability plays a prominent role in food security [32].

Farm Structure Survey of European countries indicates that several criteria can be used in measuring the size of farms. Because small and large farms are relative concepts [8], in this study, we improve on the lack of documentation of measuring farms through the composite indicators. To do this, we have developed a single synthesis index of farm structure across 28 European countries and 4 countries in the European Free Trade Association for a systemic international comparison and a re-define of small farms.

Once again, a composite indicator has no unit; different numbers can be compared together as smaller and larger. In the concept of the composite indicator of farm structure, a representative of farm size, which has been constructed through partial productivity concept in this paper, the higher values show us the better situations and the lower values show the worse situations. Therefore, when a country receives a higher value of the constructed composite indicator of farm structure, we know that the country has multi-criteria large farms because of high partial productivity. On the other hand, when a country receives a low value of the composite indicator of farm structure, we can say the country has multi-criteria small farms because of the low partial productivity. Lastly, in this study, a small farm has been defined as a farm that is not productive enough, according to its inputs and its outputs.

With regard to the role of multi-criteria small farms in producing food (89 percent) in Europe and the effect of farm structure on food security (positive elasticity), they are the main part of the solution to achieving and improving food security. So, to achieve food security through multi-criteria small farms, an understanding and recognition of the typology of the multi-criteria small farms must be noticed. Due to the low partial productivities of agricultural land and labor of multi-criteria small farms compared with multi-criteria large farms, to improve the partial productivities of agricultural land and labor across Europe, it would be useful to define some patterns. The Netherlands can be a good pattern to use in increasing the partial productivity of land to learn how they can produce 20 tons in hectares considering the different conditions in producing different agricultural and livestock products. Or else Denmark can be defined as a useful pattern to compare the countries for improving their partial productivity of agricultural labor by producing 436 tons in annual work units. The patterns may help decision-makers in making appropriate decisions and innovations.

Appropriate interventions should be proposed for different types of multi-criteria small farms. To recognize appropriate innovations and confirm it statistically, we have drawn a scatterplot matrix (reported in Appendix A, Figure A1) to see the relation of the original indicators, i.e., agricultural land, agricultural labor, livestock, and standard output, with the final constructed composite indicator of the farm structure. For example, if we consider the relationship between farm structure and land, we can determine different countries with the same amount of land have received a different level of farm structure (small and large). Therefore, the land is not the only criterion that can be used for measuring farms and improving farm structure. On the other hand, by looking at the relationship between the livestock and standard output with farm structure, it shows that any innovation that is caused to improve the partial productivity of a farm through increasing the production of the primary crops and animals will improve the farm structure. These relationships, which have been confirmed by a Pool-OLS regression (Table A3 in Appendix A), can help decision-makers determine appropriate interventions and innovations in multi-criteria small farms.

According to our (mathematically, statistically and theoretically) findings in the current study, small farms are the farms that are not productive enough. Institutional and technical measures are needed to boost agricultural productivity. Many factors may contribute to increasing agricultural production and productivity. The policy implication, a priority, should be given to the design of policies to promote small farms; this has been found to improve partial productivities.

Small farms have several options for raising land partial productivity, including switching into higher-value, labor-intensive crops and livestock, making land-improving investments and adopting more input-intensive technologies. Small farm assistance programs need to be aware of the diversity of small farm situations today and to build strategies appropriate to each. This targeting requires the development and use of small-farm classification schemes or typologies. It may also be necessary to differentiate between small farms in dynamic versus lagging regions because of the different opportunities and constraints they face. Small farms will not be able to increase their partial productivities and sustainability unless they are prepared to innovate and are supported in doing so.

Policies have an important role to play in determining the conditions under which small farms systems work and progress. Small farms may require protection through policies, which are affected

by the higher partial productivities to help in achieving higher security in food. A crucial necessity to upgrade and finance national research and extension systems targeted specifically to the needs of small farms will be needed [10]. Agricultural partial productivity is one of the key determinants. The main objective would be to increase partial productivity and resilience through diversification of the production system with a high concern for the self-provision of diverse foods with a high nutritional value. Combining increased partial productivity and resilience will require a high level of investment in research to develop productive land-labor-use systems [34] with minimal ecological risk. Small farms need appropriate international collaboration and the sharing of experiences in different regions of the world. Since Innovations and interventions [48] can have different consequences, improving food security needs an adequate (technical, social and economic) innovation [49–51] in farms.

## 5. Conclusions

Farms, as the first actors located in the food supply chains [6,52], are a key in designing the related policies and innovations [53]. Farm size has recently become debatable because it can be measured in different ways [10,53]. Determining farm size is subjective-even if all appropriate information is available [10,13]. However, there is no single measure of farm size in agriculture, and research findings may differ according to the proxy used [13]. To improve this lack of literature, in this study, we have developed the concept of composite indicators in the context of the farm structure to make a single index as a representative of farm structure that includes the criteria that can be used in measuring the size of farms. According to the findings of the study we have received that multi-criteria small farms are producing more than 80 percent of food across European countries but in comparison to the large farms, they are producing these amounts of food with much lower partial productivities of agricultural land and labor. As to the other part of the study, we have tried to see how improvement in farm structure will improve food security across Europe through a spatial regression model. According to the results of the econometric estimation we have received, improvement in farm structure in a European country can improve food security in the country, directly, and for its neighbors, indirectly. Therefore, multi-criteria small farms are a main part of the solution in achieving and improving food security, considering and understanding their diversity. New research and studies should be done for specific products and specific countries to increase agricultural partial productivity and for environmental issues too. Good patterns should be defined for the specific products to try to reach the possible high partial productivity regarding the presence of physical and environmental conditions. Public-private decisions and investments to increase partial productivities of agricultural land and labor are seen necessary. Therefore, if more well-defined policies were adopted to introduce suitable innovations to multi-criteria small farms, they could play an important role in increasing food production and not only improve food security but to be a driver of growth.

**Author Contributions:** Conceptualization, F.B. and G.B., Formal analysis, Z.A.; Funding acquisition, G.B.; Methodology, Z.A. and F.B.; Software, Z.A.; Supervision, G.B.; Validation, F.B. and G.B.; Writing – original draft, Z.A.; Writing – review & editing, F.B. and G.B. All authors have read and agreed to the published version of the manuscript.

**Funding:** The authors gratefully acknowledge funding from Small farms, Small businesses and Sustainable food and nutrition security (SALSA) project (Grant Agreement No. 677363).

**Conflicts of Interest:** The authors declare no conflict of interest.

## Appendix A

**Table A1.** The obtained weights of the indicators through factor analysis; Source: own calculations.

| Indicators | Weights |
|---|---|
| Food availability: | |
| Average dietary energy supply adequacy (percentage) | 0.160 |
| Average value of food production (Dollars) | 0.041 |
| Share of dietary energy supply derived from cereals, roots, and tubers (percentage) | 0.099 |
| Average protein supply (grams) | 0.344 |
| Average supply of protein of animal origin (grams) | 0.357 |
| Farm structure: | |
| Average area of holdings (hectare) | 0.191 |
| Average livestock units of Holdings (LSU) | 0.317 |
| Average labor force of holdings (annual work units) | 0.206 |
| Average standard output of holdings (Euros) | 0.286 |
| Climatic conditions: | |
| Temperature (centigrade) | 0.500 |
| Rainfall (millimeter) | 0.500 |
| Economic factors: | |
| Gross domestic product per capita (Dollars) | 0.500 |
| Domestic food price level index | 0.500 |

**Table A2.** The order of European countries through the composite indicator of farm structure; Source: own calculations.

| Countries | Composite Indicator of Farm Structure | Clusters |
|---|---|---|
| Belgium | 5.57 | 3 |
| Netherlands | 5.23 | 3 |
| Switzerland | 4.96 | 3 |
| Norway | 4.56 | 3 |
| Denmark | 4.49 | 3 |
| Slovenia | 4.34 | 3 |
| Austria | 3.50 | 2 |
| Luxembourg | 3.49 | 2 |
| Italy | 3.47 | 2 |
| Germany | 3.36 | 2 |
| France | 3.16 | 2 |
| Cyprus | 2.64 | 2 |
| Ireland | 3.14 | 2 |
| Croatia | 3.09 | 2 |
| Spain | 2.94 | 2 |
| United Kingdom | 2.92 | 2 |
| Poland | 2.79 | 2 |
| Finland | 2.72 | 2 |
| Sweden | 2.72 | 2 |
| Portugal | 2.65 | 2 |
| Greece | 2.14 | 1 |
| Hungary | 2.05 | 1 |
| Estonia | 1.89 | 1 |
| Slovakia | 1.86 | 1 |
| Czech Republic | 1.65 | 1 |
| Iceland | 1.36 | 1 |
| Malta | 1.34 | 1 |
| Lithuania | 1.28 | 1 |
| Latvia | 1.28 | 1 |
| Bulgaria | 1.16 | 1 |
| Montenegro | 1.15 | 1 |
| Romania | 0.23 | 1 |

3: Multi-criteria large farms, 2: Multi-criteria medium farms, 1: Multi-criteria small farms.

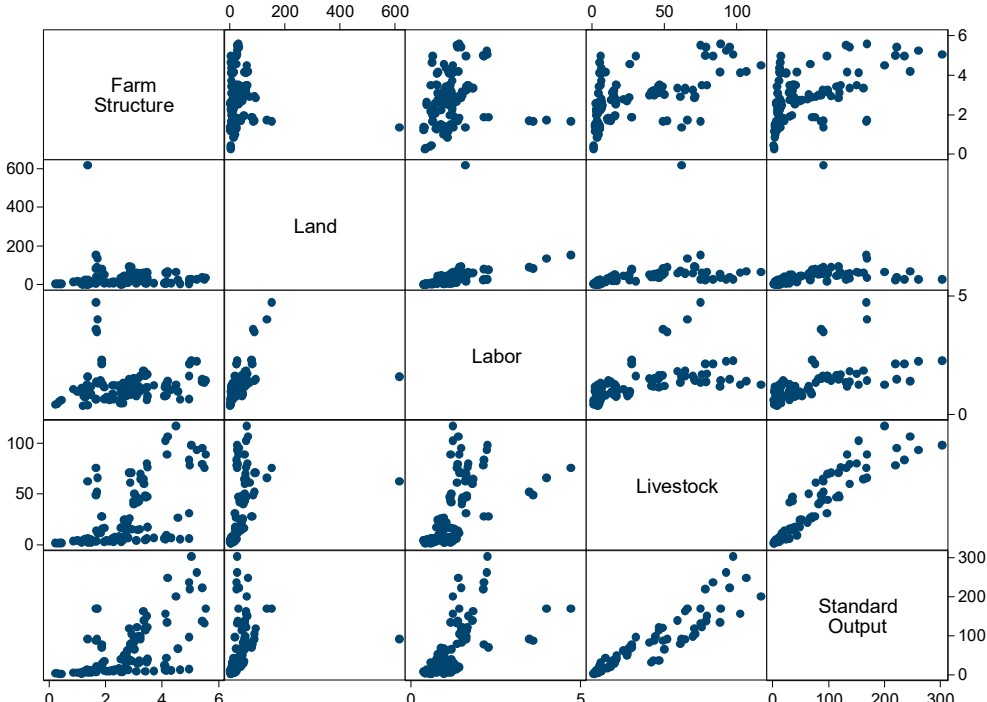

**Figure A1.** Scatterplot matrix; Farm structure: land, labor, livestock, and standard output; Source: own calculation.

**Table A3.** The relation of the composite indicator of farm structure and the original data; Source: own calculation.

| Farm Structure | Coefficient | $p$-Value | Number of Observations = 115 |
|---|---|---|---|
| Land | −0.005 | 0.001 | F (3, 111) = 33.66 |
| Labor | −0.517 | 0.002 | Prob > F = 0.000 |
| Outputs | 0.754 | 0.000 | R-squared = 0.476 |
| Intercept | 3.575 | 0.000 | Adj $R$-squared = 0.462 |

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
