# Peer review of "New Evaluation of Small Farms: Implication for an Analysis of Food Security"

_agriculture, doi:10.3390/agriculture10030074_

Round 1
Reviewer 1 Report
1. I am generally sympathetic to the research. However:
2. I miss a lot of relevant literature in your article. I suggest you discuss at least Kostov, Davidova, and Bailey 2019; Kostov, Davidova, and Bailey 2018; Davidova and Thomson 2014, Wuepper, Wimmer, and Sauer 2020, 2019, OECD 2005; Lowder, Skoet, and Raney 2016; van Vliet et al. 2015; Garner and de la O Campos 2014. Especially Wuepper et al 2020 exactly discuss how to identify different farm types.
2. The article requires language editing and a clear explanation how you chose your indicators (apparently not by refering to the main literature?)
References
1. Davidova, Sophia, and Kenneth Thomson. 2014. "Family farming in Europe: Challenges and Prospects." In Report for the European Parliament's Committee on Agriculture and Rural Development.
2. Garner, Elizabeth, and Ana Paula de la O Campos. 2014. 'Identifying the “family farm”: an informal discussion of the concepts and definitions', Food and Agriculture Organization of the United Nations (FAO), Rome Google Scholar.
3. Kostov, Philip, Sophia Davidova, and Alastair Bailey. 2019. 'Comparative efficiency of family and corporate farms: does family labour matter?', Journal of Agricultural Economics, 70: 101-15.
4. Kostov, Philip, Sophia Davidova, and Alistair Bailey. 2018. 'Effect of family labour on output of farms in selected EU Member States: a non-parametric quantile regression approach', European Review of Agricultural Economics, 45: 367-95.
5. Lowder, Sarah K, Jakob Skoet, and Terri Raney. 2016. 'The number, size, and distribution of farms, smallholder farms, and family farms worldwide', World Development, 87: 16-29.
6. OECD. 2005. "Farm Structure and Farm Characteristics - Links to Non-Commodity Outputs and Externalities." In Report prepared by the working party on agricultural policies and markets of the committee for agriculture.
7. van Vliet, Jiska A., Antonius G. T. Schut, Pytrik Reidsma, Katrien Descheemaeker, Maja Slingerland, Gerrie W. J. van de Ven, and Ken E. Giller. 2015. 'De-mystifying family farming: Features, diversity and trends across the globe', Global Food Security, 5: 11-18.
8. Wuepper, David, Stefan Wimmer, and Johannes Sauer. 2019. 'The Effect of Family Farming on Employment and Migration: Empirical Evidence from Germany', Working Paper Technical University Munich.
9. Wuepper, David, Stefan Wimmer, and Johannes Sauer. 2020. 'Is small family farming more environmentally sustainable? Evidence from a spatial regression discontinuity design in Germany', Land use policy, 90: 104360.
Reviewer 2 Report
I am fine with the current version although the scientific soundness and novelty is limited.
Reviewer 3 Report
The article is generally interesting and raises an important issue. The problem of defining small farms is difficult and causes that the results of various studies are not comparable. That is why I am glad that the authors have taken up this issue. However, my expectations were not fully met and my initial enthusiasm after reading the articles weakened.
I expected different content after reading the title. It suggests that the use of different measures of farm size affects the measurement of food security.
I rate the introduction section quite positively.
However, the literature section needs improvement. It is basically limited to discussing the composite indicator issue. You can eliminate it and move some of the text to the introduction section and some to the methods section. However, I would suggest supplementing it and discussing on the basis of literature why these and not other indicators were chosen for the composite index construction. For example: it is not clear to me why the LSU indicator was chosen? After all, not all farms have to have animals. In fact, the largest specialized cereal farms do not have them. I also don't understand why indicator temperature was chosen. What is its impact on food security? Neither too low nor too high is good, so it cannot be used that way to construct the index used in the regression. A similar problem is with the impact of the price index. Your results suggest that rising prices are conducive to food security. But to a certain level. In countries with high inflation, there are problems with access to food. These doubts should be discussed and explained based on the literature review.
Lines 349-358 - it is not clear. How the "... the composite indicator of farm structure, has been constructed through partial productivity concept...".
I do not like the discussion and conclusions section. Some conclusions do not result from research and article. These are colloquial views. For example: lines 384-389 - why do you write about management? You did not analyse it in your paper.
It looks like the article was earlier longer and was shortened but with not the best result.
Numbers by references are also confused. I do not think that items such as 27 and 37 discuss the issues assigned to them in the article.
In addition, it would be good to give an article for professional proofreading. Some confusing statements may result from incorrectly chosen words and wrong sentence structures.
Round 2
Reviewer 1 Report
Dear authors,
I think your paper has much improved.
Author Response
Dear Reviewer,
We appreciate for the positive comment about the current version of our manuscript.
Sincerely,
Authors
Reviewer 3 Report
I am not entirely satisfied with the authors' answers and corrections.
Point 1: I understand what you did in the article, but the title indicates a methodological approach. I suggest you modify it.
Point 2: Your answers don't convince me. Just because someone else used some variables (even if it is FAO and Eurostat) does not mean that they can be used in any case. Anyway, these institutions also use other variables to describe the level and dimensions of food security and farm structure. Depends on the study and context. You can't just write that you get FAO and Eurostat data. If you want to use them, you need to better explain the set of variables based on the literature review and justify its the selection.
2.2 and point 3: The explanation about the construction of the index using the concept of productivity should be included in the text (not just in reviewer's response). The design of the indicator is not sufficiently described. Besides, in the file you sent, there are no lines 250-258, to which you refer.
2.3 In response, you actually confirm my complaint. Precipitation and temperature have different effects on food security. Therefore, they cannot be used as a stimulant in regression. You take them as stimulants in regression, so variables working in one direction, which is not true (e.g. rainfall).
Point 4:You have only corrected / deleted the part related to management. My comment related to the whole conclusions. I just gave you one example.
Point 5: Yes, the article is not consistent. There are logical abbreviations, some issues are not sufficiently clarified, which gives the impression that fragments have been cut out of it.
